# Association between Preoperative Nutritional Status and Clinical Outcomes of Patients with Coronary Artery Disease Undergoing Percutaneous Coronary Intervention

**DOI:** 10.3390/nu12051295

**Published:** 2020-05-02

**Authors:** Su-Chan Chen, Ya-Ling Yang, Cheng-Hsueh Wu, Shao-Sung Huang, Wan Leong Chan, Shing-Jong Lin, Chia-Yu Chou, Jaw-Wen Chen, Ju-Pin Pan, Min-Ji Charng, Ying-Hwa Chen, Tao-Cheng Wu, Tse-Min Lu, Pai-Feng Hsu, Po-Hsun Huang, Hao-Min Cheng, Chin-Chou Huang, Shih-Hsien Sung, Yenn-Jiang Lin, Hsin-Bang Leu

**Affiliations:** 1Division of Cardiology, Department of Medicine, Taipei Veterans General Hospital, Taipei 11217, Taiwan; stupine1986@gmail.com (S.-C.C.); jjbear0915@gmail.com (Y.-L.Y.); chwu6@vghtpe.gov.tw (C.-H.W.); sshuang2@vghtpe.gov.tw (S.-S.H.); wlchan@vghtpe.gov.tw (W.L.C.); sjlin@vghtpe.gov.tw (S.-J.L.); cychou@vghtpe.gov.tw (C.-Y.C.); jwchen@vghtpe.gov.tw (J.-W.C.); jppan@vghtpe.gov.tw (J.-P.P.); mjcharng@vghtpe.gov.tw (M.-J.C.); yhchen@vghtpe.gov.tw (Y.-H.C.); dcwu@vghtpe.gov.tw (T.-C.W.); tmlu@vghtpe.gov.tw (T.-M.L.); pfhsu57@gmail.com (P.-F.H.); huangbsvgh@gmail.com (P.-H.H.); circulation0913@gmail.com (H.-M.C.); cchuang4@vghtpe.gov.tw (C.-C.H.); mr.sungsh@gmail.com (S.-H.S.); linyennjiang@gmail.com (Y.-J.L.); 2Faculty of Medicine, National Yang-Ming University, Taipei 11221, Taiwan; 3Institute of Clinical Medicine and Cardiovascular Research Center, National Yang-Ming University, Taipei 11221, Taiwan; 4Department of Medical Education, Taipei Veterans General Hospital, Taipei 11217, Taiwan

**Keywords:** coronary artery disease, percutaneous coronary intervention, nutrition, risk stratification, Controlling Nutritional Status (CONUT) score

## Abstract

**Background:** Malnutrition is associated with poor outcomes in patients with cancer, heart failure and chronic kidney disease. This study aimed to investigate the predictive value of the Controlling Nutritional Status (CONUT) score in coronary artery disease (CAD) patients. Methods: We recruited a cohort of 3118 patients with CAD undergoing percutaneous coronary intervention (PCI) from 2005 to 2015. Nutritional status was evaluated using the CONUT score, with higher scores reflecting worse nutritional status. **Results:** After adjustment for comorbidities and medication, an increased CONUT score was independently associated with a higher risk of acute myocardial infarction (AMI) (HR: 1.13; 95% CI: 1.03–1.24), cardiovascular (CV) death (HR: 1.18; 95% CI: 1.07–1.30), congestive heart failure (CHF) (HR: 1.11; 95% CI: 1.04–1.18), a major adverse cardiovascular event (MACE) (HR: 1.14; 95% CI: 1.07–1.22), and total CV events (HR: 1.11; 95% CI: 1.07–1.15). The subgroup analyses demonstrated that the association of the CONUT score existed independently of other established cardiovascular risk factors. In addition, CONUT significantly improved risk stratification for myocardial infarction (MI), cardiac death, CHF, MACEs and total CV events compared to conventional risk factors in CAD patients by the significant increase in the C-index (*p* < 0.05) and reclassification risk categories in cardiac death and MACEs. **Conclusions:** The CONUT score improved the risk prediction of adverse events compared to traditional risk factors in CAD patients after percutaneous coronary intervention (PCI).

## 1. Introduction

Cardiovascular disease remains one of the leading causes of death worldwide despite improvements in evidence-based management [1,2]. Cardiovascular disease not only impairs quality of life but also imposes a heavy economic burden in many countries [3]. Apart from conventional risk factor modification, a multidisciplinary approach is necessary to achieve improved clinical outcomes and cost-effectiveness.

Malnutrition is known to be associated with worse clinical outcomes in patients with heart failure, ischemic stroke, cancer and chronic kidney disease [3,4,5,6,7]. Albumin, body weight, body mass index, cholesterol, etc. are commonly used indices to determine nutritional status but are often inaccurate. Various tools have been developed to evaluate nutritional status including the Geriatric Nutritional Risk Index (GNRI), Prognostic Nutritional Index (PNI), Controlling Nutritional Status (CONUT) score and Mini Nutritional Assessment (MNA). These nutritional assessment tools showed prognostic value in patients with malignancy, heart failure, peripheral arterial disease and coronary artery disease [8,9,10,11]. Various studies have shown PNI to be a promising means of risk stratification for stable coronary artery disease (CAD) and acute ST-elevation myocardial infarction (STEMI) patients [11,12]. However, research was lacking in the utility of the CONUT score for the purposes of assessing clinical outcomes in CAD patients. Therefore, we aim to evaluate the prognostic value of the CONUT score in patients with coronary artery disease undergoing percutaneous coronary intervention (PCI).

## 2. Materials and Methods

### 2.1. Study Population 

A total of 3118 patients with symptomatic CAD who received coronary intervention at the Taipei Veterans General Hospital between 2005 and 2015 were enrolled in this study. CAD was diagnosed by at least one of the following modalities: (1) a history of myocardial infarction as evidenced by ischemic change in a 12-lead electrocardiography (ECG) and elevated cardiac enzymes; or (2) a history of angina with ischemic ECG changes, a positive response to a stress test, or the presence of a significant stenotic lesion on a coronary computed tomography angiography (CCTA). CAD patients who fulfilled the above criteria and received a coronary intervention (percutaneous coronary intervention (PCI)) with either coronary stenting or balloon angioplasty were enrolled. This work was a retrospective observational study that complied with the Declaration of Helsinki and was approved by the appropriate Health Authorities, Independent Ethics Committees, and Independent Review Boards in Taipei Veterans General Hospital (2016-03-014CC). 

### 2.2. Baseline Characteristics and Biochemical Data

Baseline characteristics and risk factors, including a history of hypertension, diabetes and smoking, as well as medication history, were collected. In addition, biochemical profiles including albumin, blood profiles, lipid profiles, renal function parameters and parameters related to coronary intervention were collected. 

### 2.3. Nutrition Status Evaluation 

Nutritional status was evaluated using the CONUT score, which takes into account serum values of albumin (g/dL), total cholesterol level (mg/dl), and total lymphocyte count (/mL), with a score ranging from 0 to 12 [6,13]. Higher scores reflect a worse nutritional status and have been used for nutrition status evaluation [13]. Subjects were classified as normal (0–1) or mildly (2–4), moderately (5–8) or severely (9–12) malnourished based on the CONUT score. A similar classification of nutritional status has been reported in other studies [6,7].

### 2.4. Clinical Follow-Up for Future Adverse Cardiovascular Events 

The study patients resumed outpatient clinic visits within two to four weeks of discharge. After their first return visit, they were regularly followed up at one- to three-month intervals. Data for follow-up were retrospectively obtained from hospital records and chart reviews. The primary endpoint was major cardiovascular events including cardiovascular death, non-fatal myocardial infarction, non-fatal stroke and revascularization procedures including coronary intervention and bypass surgery. Myocardial infarction was confirmed in patients presenting with ischemic symptoms with elevated serum cardiac enzyme levels and/or characteristic ECG changes. Coronary revascularization procedures with either coronary intervention or coronary artery bypass grafting surgery were confirmed by medical record review. Stroke was confirmed if there was a new neurologic deficit lasting for at least 24 h with definite imaging evidence of cerebrovascular accident by either MRI or CT scan. Major adverse cardiovascular events (MACEs) included cardiac death, non-fatal myocardial infarction (MI) and ischemic stroke. The protocol for cardiovascular (CV) event follow-up was similar to that previously reported [14,15]. 

### 2.5. Statistical Analyses 

The baseline characteristics of patients according to baseline CONUT score categories were compared. The occurrence of future adverse outcomes including non-fatal stroke, non-fatal myocardial infarction, repeat revascularization, and total CV events during the follow-up period was compared between groups. Quantitative variables were expressed as the mean and standard deviation in the presence of a normal distribution, or as the median and interquartile range in the presence of an asymmetric distribution. Qualitative variables were presented as both absolute frequencies (number of patients) and relative frequencies (percentage). Comparisons of continuous variables between groups were performed by an ANOVA test, while subgroup comparisons of categorical variables were assessed by a χ^2^ or Fisher’s exact test. The primary and secondary outcomes were described as overall percentages and expressed as means of proportions with a 95% confidence interval (CI). The event-free survival rate was calculated using the Kaplan–Meier method, with the significance evaluation using log rank tests. The primary analysis used an unstratified log-rank test to compare the overall survival between variable CONUT groups. Multiple regression analysis was carried out using Cox proportional hazard regression analysis adjusted for age, gender, BMI, history of hypertension, diabetes, medications including statins, LDL and HDL to evaluate whether the CONUT score was an independent factor in determining the occurrence of acute myocardial infarction, congestive heart failure (CHF), CV death, major cardiovascular events, and total CV events. Subsequent subgroup analysis was performed to investigate the effects of the CONUT score among other risk factors for cardiovascular events, such as age, gender, history of diabetes mellitus, hypertension, LDL and HDL. To assess whether the accuracy of predicting adverse cardiac events would improve after the addition of CONUT to a baseline model with established risk factors (i.e., age, gender, hypertension, diabetes, BMI and medications), the C-index, net reclassification improvement (NRI), and integrated discrimination improvement (IDI) were calculated. A similar analysis has been used in our previous work [16]. Statistical analysis was performed utilizing the SPSS software (Version 15.0, IBM Corporation, Armonk, NY, USA) and R version 3.2.3 (http://www.R-project.org/; R Foundation for Statistical Computing, Vienna, Austria). In all of the tests, the two-tailed alpha significance level was 0.05. 

## 3. Results

### 3.1. Baseline Characteristics

A total of 3118 patients who underwent percutaneous coronary intervention were enrolled in this study. The baseline characteristics of the participants according to CONUT score categories are shown in Table 1. The mean age of our patients was 71.5 ± 12.1 years, of whom 81.5% were male, and the mean CONUT score was 2.6. Patients with higher CONUT score were older, had lower body weights, lower BMIs, lower LDL-C values and worse renal function (*p* for trend < 0.05). In addition, patients with worse nutrition had more underlying comorbidities including congestive heart failure, chronic kidney disease and more extensive coronary artery disease. 

### 3.2. Association of Nutritional Status and Clinical Outcomes of Patients After Percutaneous Coronary Intervention

Table 2 shows the clinical outcomes according to baseline CONUT score categories. During a follow-up of 58.5 ± 35.8 months in the entire cohort, there were 197 acute myocardial infarctions (6.3%), 109 cardiac deaths (3.5%), 366 congestive heart failures (11.7%), 583 revascularizations (18.7%), 370 MACEs (11.9%) and 1098 total CV events (35.2%). Poor nutrition with increased baseline CONUT categories was correlated with a higher incidence of acute myocardial infarction, cardiac death, congestive heart failure, revascularization, MACEs and total CV events (Table 2). Figure 1 shows the relationship between baseline CONUT categories and the future risk of an event according to Kaplan–Meier survival analysis. The Kaplan–Meier analysis showed that patients with a high CONUT score were significantly associated with higher rates of major events including MACEs (log-rank *p* < 0.001), AMI (log-rank *p* < 0.001), cardiovascular death (log-rank *p* < 0.001), CHF (log-rank *p* < 0.001) and total CV events (log-rank *p* < 0.001). This suggests that a higher CONUT category (poor nutrition status) was associated with an increased risk of future adverse events in CAD patients after coronary intervention.

Cox regression analysis further confirmed the independent predictive role of the CONUT score in the future risk of an adverse cardiovascular event after adjustment for age, gender, BMI, lipid profile, renal function and medication (Table 3). An increasing CONUT score was independently associated with a higher risk of acute myocardial infarction (AMI) (HR: 1.13; 95% CI: 1.03–1.24; *p* = 0.008), CV death (HR: 1.18; 95% CI: 1.07–1.30; *p* = 0.001), CHF (HR: 1.11; 95% CI: 1.04–1.18; *p* = 0.002), MACEs (HR: 1.14; 95% CI: 1.07–1.22; *p* < 0.001) and total CV events (HR: 1.11; 95% CI: 1.07–1.15; *p* < 0.001). The trends toward increased risks of AMI, CV death, CHF, MACEs and total CV events were significant across worse CONUT categories (Table 3). Both showed that higher CONUT scores are associated with worse outcomes in patients after coronary intervention. The subgroup analyses further demonstrated that the association of CONUT score existed independently of other established cardiovascular risk factors including gender group, age, history of hypertension, diabetes, smoking status, the presence of myocardial infarction, BMI, lipid profile, baseline renal function, heart function and statin use. This indicates that the CONUT score is a useful independent marker for risk stratification in patients with CAD undergoing PCI (Appendix A).

### 3.3. Additional Predictive Values after Considering CONUT in Predicting Future Risk in CAD Patients after PCI

Adding the CONUT score to a baseline model improved the prediction of cardiac death (*p* = 0.0062), nonfatal MI (*p* = 0.0483), CHF (*p* = 0.001), MACEs (*p* < 0.001) and total CV events (*p* = 0.0053), as shown by the significant increase in the C-index (Table 4). Reclassification through the addition of the CONUT score also showed a significant integrated discrimination improvement (IDI) of 0.0135 (*p* < 0.001) with a 12.29% increase in net reclassification improvement (NRI) (*p* = 0.0151) in cardiac death and MACEs (NRI: 0.0244, *p* = 0.0481; IDI: 0.0035, *p* = 0.039). This suggests that adding the CONUT score could provide significantly better predictive value than traditional risk factors for CAD patients after PCI, especially for cardiac death and major adverse events.

## 4. Discussion

Our present study showed that patients with higher CONUT scores had an increased risk of cardiac death, nonfatal MI, congestive heart failure and major adverse cardiovascular events (MACEs), suggesting that CONUT independently represents a useful predictive value regarding the long-term outcomes of CAD patients. Subgroup analyses emphasized that the CONUT score is an independent marker for risk stratification after adjusting for other cardiovascular risk factors in high-risk CAD populations. In addition, the CONUT score significantly improved the risk stratification of cardiac death and MACEs compared to conventional risk factors in CAD patients. This indicates that the CONUT could be used as an easy and practical indicator for identifying high-risk CAD patients who will develop major events after PCI.

Nutrition status plays an important role in maintaining vital organ function in the human body [17,18]. Malnutrition is associated with worse outcomes in the elderly, critically ill patients, chronic kidney disease, heart failure and ischemic stroke [5,19,20,21]. Although nutrition status is very important, the correlation between malnutrition and cardiovascular disease has focused mainly on the CHF population. Our previous study demonstrated that a low serum albumin concentration worsens the prognosis of patients with stable CAD [22], suggesting the important role that nutrition status plays in determining the long-term outcomes in CAD patients. 

The CONUT score consists of three laboratory markers: albumin, cholesterol and lymphocyte count. Albumin levels are not related solely to nutritional status. They are also related to the acute phase reaction. Albumin is a serologic marker of inflammation superimposed on malnutrition, which has inflammatory effects on the vascular endothelium and lipoprotein structure [23]. Lower albumin levels may be a marker of persistent injury to the arteries and the progression of atherosclerosis and thrombosis [24]. Cholesterol homeostasis occurs as part of innate immune response and its disruption may augment inflammatory responses, leading to atherosclerosis [23,25]. The link between elevated serum cholesterol levels and cardiovascular disease was confirmed with the discoveries of the low-density lipoprotein (LDL) receptor and statins [26]. However, the lipid paradox has been reported in several clinical studies including myocardial infarction and heart failure patients [27]. Though the cause of the lipid paradox remains undetermined, possible explanations are that these patients have a high vascular inflammation status even though their LDL is not very high. Low lipid levels may be markers of advanced disease and systemic inflammatory activation [28], and could be a reflection of malnutrition and cachexia, which are known to be associated with increased mortality in different chronic diseases [5,19,20,21]. Lower LDL levels may be associated with poor nutritional status, making patients more vulnerable to future events. Furthermore, these patients may not receive aggressive statin therapy because of their lower baseline LDL levels, and relatively low statin use may be responsible for the higher risk in the high CONUT score group. To elucidate this issue, our current study adjusted for confounding factors including statin use, LDL and HDL values, and a higher CONUT score was still independently associated with the risk of a future adverse event in CAD patients after PCI. Lymphocyte counts reflect the host immunity and have been studied with respect to their association with nutritional status [29]. Lymphocytopenia may reflect neurohormonal activation in CAD patients and may represent a marker of the physiological stress response [30,31]. A low lymphocyte percentage was shown to be independently and significantly associated with CAD and could be used as a predictor of worse prognosis [32]. These elements contribute to the pathophysiology of coronary artery disease [33]. This makes the CONUT score a reasonable tool for determining the nutritional status of the CAD population. In previous studies, the CONUT score was shown to predict outcomes in populations with acute heart failure [8], acute ischemic stroke [6], elderly hypertension [34], and cancers including mesothelioma and gastric cancer [7,35]. 

Recently H. Wada et al. disclosed that nutritional status as assessed by the PNI is useful in predicting long-term cardiovascular outcomes in stable CAD patients that underwent elective PCI [11]. In this study, the lowest PNI group was significantly associated with MACEs and all-cause death compared to the highest PNI group even after adjustments for other risk factors. Our study showed similar results with nutritional evaluation using the CONUT score, but our cohort included patients presenting with acute coronary syndromes. Furthermore, our study demonstrated that an increased CONUT score was independently associated with an increased future risk of AMI, CV death, CHF, MACEs and total CV events, extending the predictive value of nutrition scoring in CAD patients after PCI. 

The mechanism linking poor nutrition to worse adverse outcomes is considered to be multifactorial. Malnutrition may decrease underlying fibrinolysis ability, platelet inhibition and antioxidant capacity and increase blood viscosity, leading to the occurrence of adverse outcomes. Our study is consistent with the finding of M. Yokoyama who reported the independent predictive value of CONUT score-based malnutrition status in peripheral arterial disease [36]. In addition, the subgroup analyses showed that the association of the CONUT score existed independently of other established cardiovascular risk factors, especially among these high-risk patients. This suggests that the CONUT score is an independent prognosis predictor in CAD patients. Furthermore, the addition of the CONUT score to the clinical model improved not only the predictive power for major cardiovascular events such as cardiac death and major adverse cardiac events (as assessed by the receiver operating characteristic (ROC) curves) but also the reclassification of the subjects into different risk categories via the IDI and NRI. This implies that a high CONUT score is an independent prognostic marker of future adverse events and increases the predictive value for adverse events among CAD patients.

This study has several potential limitations due to its retrospective nature and patient enrollment at a single institution. The study period is quite long, and the treatment of coronary artery disease including the statin intensity and prescription practice has changed during this period. Additionally, due to a lack of data, we could not determine the effects of the serial changes in nutritional status. However, we showed that malnutrition is common in patients with coronary artery disease, which could add as a prognostic factor for clinical outcomes in addition to the established risk factors. The use of the CONUT score, which comprises cholesterol levels, might be affected by the use of statins, which is common in patients with coronary artery disease. This issue has been adjusted for by considering statin use in the Cox regression analysis in Model 2 and Model 3 as shown in Table 3. In addition, the subgroup analysis of statin use shows an independent association of the CONUT score and future CV events in patients after PCI (as demonstrated in Appendix A). Finally, the role of nutritional support was well established in surgical patients [37]. Nutritional support was associated with fewer complications and a shorter length of stay in patients at nutritional risk [38]. However, there was lack of research regarding the impact of nutrition therapy in patients with coronary artery disease. This can be a focus of attention for future research.

## 5. Conclusions

This study shows that the CONUT score is an independent predictor of long-term cardiovascular disease to identify outcomes in patients with coronary artery disease, including those presenting with acute coronary syndromes. This suggests the usefulness of the CONUT score, in addition to the well-established cardiovascular risk factors, for risk stratification in these patients prior to PCI. 

## Figures and Tables

**Figure 1 nutrients-12-01295-f001:**
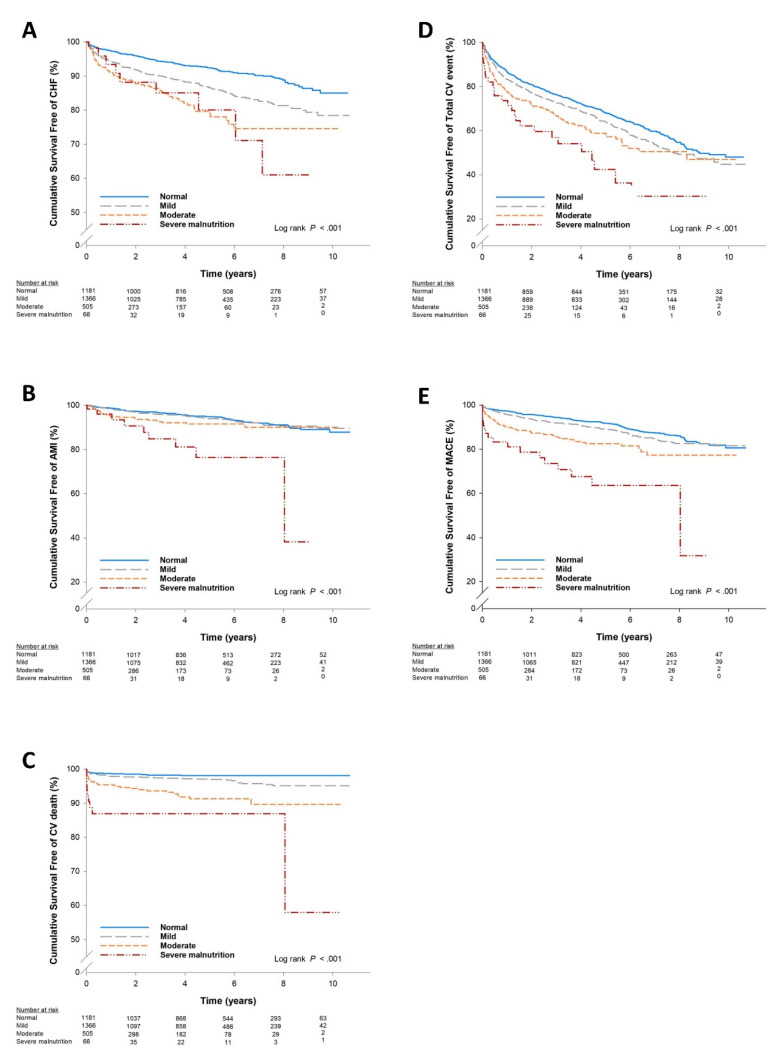
Cumulative survival free of events stratified by different nutritional categories based on Kaplan–Meier analysis: (**A**) congestive heart failure (CHF); (**B**) acute myocardial infarction (AMI); (**C**) cardiovascular death; (**D**) total cardiovascular events; (**E**) major adverse cardiovascular events (MACEs).

**Table 1 nutrients-12-01295-t001:** Baseline characteristics of patients in different categorical groups of nutritional status.

Baseline Characteristics	Overall (*n* = 3118)	Nutritional Status as Classified by CONUT				
		Normal (*n* = 1181)	Mild (*n* = 1366)	Moderate (*n* = 505)	Severe (*n* = 66)	*p* Value
Age, years	71.50 ± 12.12	68.17 ± 12.67	73.01 ± 11.31	74.71 ± 11.49	75.11 ± 8.83	<0.001
Male, *n* (%)	2542 (81.5)	947 (80.2)	1153 (84.4)	390 (77.2)	52 (78.8)	0.001
SBP, mm Hg	128.47 ± 19.54	128.75 ± 19.09	128.98 ± 19.43	126.54 ± 20.72	127.45 ± 20.46	0.117
DBP, mm Hg	70.52 ± 11.43	72.37 ± 11.32	70.06 ± 11.29	67.90 ± 11.30	66.57 ± 11.79	<0.001
Height, cm	163.66 ± 8.13	164.30 ± 8.36	163.81 ± 7.76	162.53 ± 8.22	161.44 ± 9.51	0.001
Weight, kg	66.69 ± 12.07	70.35 ± 12.10	66.27 ± 11.39	62.34 ± 11.29	59.14 ± 13.01	<0.001
BMI, kg/m^2^	24.78 ± 3.68	25.94 ± 3.57	24.57 ± 3.44	23.53 ± 3.76	22.56 ± 3.86	<0.001
Smoking, *n* (%)	1272 (40.8)	473 (40.1)	551 (40.3)	217 (43)	31 (47)	0.490
**Medical history**						
Acute coronary syndrome, *n* (%)	1323 (42.4)	408 (34.5)	559 (40.9)	305 (60.4)	51 (71.33)	<0.001
Hypertension, *n* (%)	1980 (63.5)	768 (65)	908 (66.5)	273 (54.1)	31 (47)	<0.001
Diabetes mellitus, *n* (%)	1133 (36.3)	440 (37.3)	497 (36.4)	171 (33.9)	25 (37.9)	0.607
Dyslipidemia, *n* (%)	375 (12)	105 (8.9)	178 (13)	87 (17.2)	5 (7.6)	<0.001
Chronic kidney disease, *n* (%)	264 (8.5)	41 (3.5)	115 (8.4)	89 (17.6)	19 (28.8)	<0.001
Congestive heart failure, *n* (%)	549 (17.6)	148 (12.5)	251 (18.4)	136 (26.9)	14 (21.2)	<0.001
Stroke, *n* (%)	233 (7.5)	73 (6.2)	116 (8.5)	38 (7.5)	6 (9.1)	0.161
Peripheral artery disease, *n* (%)	212 (6.8)	60 (5.1)	102 (7.5)	45 (8.9)	5 (7.6)	0.018
Malignancy, *n* (%)	130 (4.2)	42 (3.6)	56 (4.1)	27 (5.3)	5 (7.6)	0.187
**Medication**						
Aspirin, *n* (%)	2287 (73.3)	883 (74.8)	1012 (74.1)	356 (70.5)	36 (54.5)	0.001
P_2_Y_12_ inhibitor, *n* (%)	795 (25.5)	295 (25)	341 (25)	131 (25.9)	28 (42.4)	0.001
ACE inhibitors, *n* (%)	586 (18.8)	212 (18)	269 (19.7)	96 (19)	9 (13.6)	0.487
ARBs, *n* (%)	1146 (36.8)	445 (37.7)	521 (38.1)	163 (32.3)	17 (25.8)	0.025
Beta blockers, *n* (%)	1368 (43.9)	525 (44.5)	607 (44.4)	212 (42)	24 (36.4)	0.460
Calcium channel blockers, *n* (%)	1045 (33.5)	408 (34.5)	475 (34.8)	150 (29.7)	12 (18.2)	0.008
Statins, *n* (%)	1580 (50.7)	649 (55)	678 (49.6)	232 (45.9)	21 (31.8)	<0.001
**Lab**						
Creatinine, mg/dl	1.80 ± 1.96	1.38 ± 1.32	1.89 ± 2.03	2.41 ± 2.60	2.71 ± 2.72	<0.001
eGFR, ml/min/1.73 m2	57.32 ± 27.03	64.63 ± 24.73	55.38 ± 26.58	47.19 ± 28.38	43.95 ± 28.46	<0.001
Glucose, mg/dL	122.49 ± 43.90	122.79 ± 42.97	120.94 ± 42.28	127.09 ± 51.67	117.93 ± 33.71	0.138
HbA1c, %	7.05 ± 1.39	7.12 ± 1.42	6.96 ± 1.35	7.09 ± 1.41	7.02 ± 1.29	0.122
Uric acid, mg/dl	6.55 ± 2.01	6.48 ± 1.79	6.57 ± 2.02	6.69 ± 2.37	6.36 ± 2.54	0.208
Total bilirubin, mg/dl	0.72 ± 0.63	0.72 ± 0.67	0.72 ± 0.61	0.70 ± 0.60	0.71 ± 0.56	0.960
Cholesterol, mg/dl	170.11 ± 40.64	191.97 ± 34.32	161.98 ± 37.62	146.45 ± 37.68	128.33 ± 29.93	<0.001
Triglyceride, mg/dl	132.32 ± 86.16	159.27 ± 98.53	121.21 ± 76.40	103.82 ± 62.65	95.35 ± 54.98	<0.001
HDL-C, mg/dl	42.49 ± 12.25	43.39 ± 11.83	42.99 ± 12.59	39.56 ± 11.77	37.32 ± 11.83	<0.001
LDL-C, mg/dl	105.05 ± 33.97	119.66 ± 32.33	99.53 ± 31.80	88.78 ± 29.69	76.03 ± 29.49	<0.001
Chol/HDL Ratio	4.28 ± 1.50	4.72 ± 1.49	4.02 ± 1.36	3.99 ± 1.65	3.70 ± 1.34	<0.001
Non-HDL, mg/dl	127.80 ± 39.94	148.54 ± 35.16	118.88 ± 36.79	107.16 ± 37.02	89.58 ± 28.52	<0.001
**Stent**						
Stent number, *n*	1.83 ± 1.06	1.78 ± 1.03	1.82 ± 1.03	1.97 ± 1.12	2.07 ± 1.45	0.005
Stent diameter, mm	3.08 ± 0.44	3.09 ± 0.43	3.08 ± 0.47	3.04 ± 0.42	3.13 ± 0.47	0.232
Total stent length, mm	23.36 ± 6.33	23.36 ± 5.99	23.17 ± 6.66	23.88 ± 6.36	23.48 ± 5.08	0.275
**Number of culprit vessel**						
SVD, *n* (%)	791 (25.4)	361 (30.6)	333 (24.4)	84 (16.6)	13 (19.7)	
DVD, *n* (%)	995 (31.9)	386 (32.7)	435 (318)	159 (31.5)	15 (22.7)	<0.001
TVD, *n* (%)	1329 (42.6)	433 (36.7)	596 (43.6)	262 (51.9)	38 (57.6)	

SBP, systolic blood pressure; DBP, diastolic blood pressure; ACE, angiotensin converting enzyme; ARB, angiotensin receptor blocker; CONUT, Controlling Nutritional Status; DPP-4, dipeptidyl peptidase-4; eGFR, estimated glomerular filtration rate; HbA1C, hemoglobin A1c; HDL-C, high density lipoprotein cholesterol; LDL-C, low density lipoprotein cholesterol; WBC, white blood cells; DES, drug-eluting stent; BMS, bare metal stent; SVD, single vessel disease; DVD, double vessel disease; TVD, triple vessel disease.

**Table 2 nutrients-12-01295-t002:** Clinical outcomes of patients with different nutritional status after percutaneous coronary intervention.

	Overall (*n* = 3118)	Nutritional Status as Classified by CONUT				
		Normal (*n* = 1181)	Mild (*n* = 1366)	Moderate (*n* = 505)	Severe (*n* = 66)	*p* Value
Acute myocardial infarction	197 (6.3)	77 (6.5)	80 (5.9)	31 (6.1)	9 (13.6)	0.087
Cardiovascular death	109 (3.5)	21 (1.8)	45 (3.3)	34 (6.7)	9 (13.6)	<0.001
Congestive heart failure	366 (11.7)	104 (8.8)	181 (13.3)	72 (14.3)	9 (13.6)	0.001
MACEs	370 (11.9)	126 (10.7)	155 (11.3)	71 (14.1)	18 (27.3)	<0.001
Total CV events	1098 (35.2)	414 (35.1)	493 (36.1)	163 (32.3)	28 (42.4)	0.274

MACEs, major adverse cardiovascular events; CV, cardiovascular.

**Table 3 nutrients-12-01295-t003:** Association of CONUT nutritional score and the clinical outcomes in different models.

	Crude		Model 1		Model 2		Model 3	
	HR (95% CI)	*p* Value	HR (95% CI)	*p* Value	HR (95% CI)	*p* Value	HR (95% CI)	*p* Value
**Acute myocardial infarction**								
CONUT as continuous variable	1.14 (1.07–1.21)	<0.001	1.13 (1.07–1.21)	<0.001	1.13 (1.03–1.23)	0.006	1.13 (1.03–1.24)	0.008
CONUT as categorical variable								
Normal	1.00 (reference)	-	1.00 (reference)	-	1.00 (reference)	-	1.00 (reference)	-
Mild	1.02 (0.75–1.40)	0.896	0.99 (0.72–1.36)	0.938	0.98 (0.60–1.59)	0.924	1.02 (0.62–1.70)	0.926
Moderate	1.61 (1.06–2.45)	0.026	1.55 (1.01–2.38)	0.044	1.33 (0.73–2.41)	0.356	1.36 (0.73–2.56)	0.336
Severe malnutrition	4.29 (2.14–8.57)	<0.001	4.14 (2.06–8.30)	<0.001	3.73 (1.52–9.15)	0.004	3.64 (1.40–9.51)	0.008
**Cardiovascular death**								
CONUT as continuous variable	1.30 (1.21–1.39)	<0.001	1.27 (1.19–1.36)	<0.001	1.16 (1.07–1.26)	<0.001	1.18 (1.07–1.30)	0.001
CONUT as categorical variable								
Normal	1.00 (reference)	-	1.00 (reference)	-	1.00 (reference)	-	1.00 (reference)	-
Mild	1.99 (1.18–3.34)	0.009	1.80 (1.07–3.05)	0.027	1.21 (0.67–2.19)	0.536	1.10 (0.89–2.05)	0.770
Moderate	4.94 (2.85–8.54)	<0.001	4.12 (2.36–7.19)	<0.001	2.49 (1.35–4.61)	0.004	2.51 (1.29–4.90)	0.007
Severe malnutrition	10.83 (4.95–23.69)	<0.001	9.18 (4.18–20.17)	<0.001	3.24 (1.18–8.90)	0.022	3.79 (1.31–10.95)	0.014
**Congestive heart failure**								
CONUT as continuous variable	1.16 (1.11–1.21)	<0.001	1.12 (1.07–1.17)	<0.001	1.11 (1.04–1.17)	0.001	1.11 (1.04–1.18)	0.002
CONUT as categorical variable								
Normal	1.00 (reference)	-	1.00 (reference)	-	1.00 (reference)	-	1.00 (reference)	-
Mild	1.73 (1.36–2.20)	<0.001	1.48 (1.16–1.89)	0.002	1.78 (1.24–2.54)	0.002	1.74 (1.21–2.52)	0.003
Moderate	2.69 (1.99–3.64)	<0.001	2.12 (1.55–2.88)	<0.001	2.24 (1.48–3.41)	<0.001	2.21 (1.42–3.45)	<0.001
Severe malnutrition	2.88 (1.46–5.70)	0.002	2.38 (1.20–4.71)	0.013	2.04 (0.80–5.18)	0.136	2.10 (0.81–5.43)	0.127
**MACE**								
CONUT as continuous variable	1.17 (1.12–1.22)	<0.001	1.16 (1.11–1.21)	<0.001	1.14 (1.07–1.20)	<0.001	1.14 (1.07–1.22)	<0.001
CONUT as categorical variable								
Normal	1.00 (reference)	-	1.00 (reference)	-	1.00 (reference)	-	1.00 (reference)	-
Mild	1.20 (0.95–1.52)	0.131	1.13 (0.89–1.43)	0.320	1.05 (0.74–1.47)	0.800	1.04 (0.72–1.48)	0.849
Moderate	2.12 (1.58–2.85)	<0.001	1.96 (1.46–2.64)	<0.001	1.82 (1.24–2.67)	0.002	1.83 (1.21–2.78)	0.005
Severe malnutrition	4.80 (2.93–7.89)	<0.001	4.47 (2.72–7.35)	<0.001	3.14 (1.63–6.05)	0.001	3.34 (1.67–6.67)	0.001
**Total CV events**								
CONUT as continuous variable	1.09 (1.06–1.11)	<0.001	1.08 (1.05–1.11)	<0.001	1.10 (1.06–1.14)	<0.001	1.11 (1.07–1.15)	<0.001
CONUT as categorical variable								
Normal	1.00 (reference)	-	1.00 (reference)	-	1.00 (reference)	-	1.00 (reference)	-
Mild	1.16 (1.02–1.32)	0.028	1.11 (0.97–1.27)	0.127	1.34 (1.10–1.62)	0.003	1.36 (1.11–1.66)	0.003
Moderate	1.46 (1.21–1.75)	<0.001	1.38 (1.15–1.67)	0.001	1.58 (1.23–2.01)	<0.001	1.62 (1.24–2.10)	<0.001
Severe malnutrition	2.26 (1.54–3.31)	<0.001	2.17 (1.48–3.18)	<0.001	2.52 (1.56–4.08)	<0.001	2.72 (1.65–4.50)	<0.001

Model 1: adjusted for age and sex. Model 2: adjusted for age, sex, body mass index, diabetes, hypertension and statins. Model 3: adjusted for age, sex, body mass index, diabetes, hypertension, statins, LDL and HDL.

**Table 4 nutrients-12-01295-t004:** Improvement in discrimination performance and calibration for risk prediction of cardiovascular events in the multivariate-adjusted model after including CONUT score.

	C-index (95% CI)	*p* Value	NRI (95% CI)	*p* Value	IDI (95% CI)	*p* Value
AMI						
Traditional risks (Age, Gender, HTN, DM)	0.6029 (0.5628–0.6430)	Ref.		Ref.		Ref.
Traditional risks + CONUT	0.6239 (0.5820–0.6658)	0.0483	0.0295 (−0.0031-0.0621)	0.0762	0.0005 (−0.0003–0.0014)	0.2318
CHF						
Traditional risks (Age, Gender, HTN, DM)	0.6735 (0.6463–0.7007)	Ref.		Ref.		Ref.
Traditional risks + CONUT	0.6891 (0.6629–0.7152)	0.0010	0.0383 (0.0108–0.0657)	0.0064	0.0003 (−0.0008–0.0013)	0.6363
CV death						
Traditional risks (Age, Gender, HTN, DM)	0.6846 (0.6329–0.7363)	Ref.		Ref.		Ref.
Traditional risks + CONUT	0.7275 (0.6786–0.7764)	0.0062	0.1229 (0.0238–0.2219)	0.0151	0.0135 (0.0057–0.0212)	0.0007
MACE						
Traditional risks (Age, Gender, HTN, DM)	0.5970 (0.5674–0.6267)	Ref.		Ref.		Ref.
Traditional risks + CONUT	0.6363 (0.6064–0.6662)	0.0001	0.0244 (0.0002–0.0487)	0.0481	0.0035 (0.0011–0.0058)	0.0039
Total CV events						
Traditional risks (Age, Gender, HTN, DM)	0.5544 (0.5362–0.5725)	Ref.		Ref.		Ref.
Traditional risks + CONUT	0.5703 (0.5522–0.5884)	0.0053	0.0011 (−0.0061–0.0082)	0.7697	0.0000 (−0.0001–0.0001)	0.8487

NRI, net reclassification improvement; IDI, integrated discrimination improvement; AMI, acute myocardial infarction; HTN, hypertension; DM, diabetes mellitus; CHF, congestive heart failure; CV, cardiovascular; MACE, major adverse cardiovascular event.

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
