# Peer review of "Association between Preoperative Nutritional Status and Clinical Outcomes of Patients with Coronary Artery Disease Undergoing Percutaneous Coronary Intervention"

_nutrients, 2020, doi:10.3390/nu12051295_

Round 1
Reviewer 1 Report
The authors describe the potential association between a higher CONUT score and worse CV outcome in a CAD population including ACS patients.
Although the study is well performed, there are several limitations:
-The study is retrospective, single center and the study period is very long. This should be mentioned in the section limitations. During the period 2005-2015 the treatment for coronary artery disease has changed a lot including the prescription pattern as well as the frequency and the intensity of statin treatment. There are numerous potential confounding factors.
-As statin treatment improves CV outcome but also increases CONUT score, this interaction should be more explored by the authors. It is not clear to what extent the score is negatively influenced by prescribing statins.
The authors should also investigate whether there is a supplementary value of the CONUT score (which refects not only nutrition but also inflammation and immune response) in comparison to a predominantly nutritional value such as albumin.
Furthermore Figure 2 is impossible to read.
Finally, the conclusion is grammatically not correct.
Author Response
- The study is retrospective, single center and the study period is very long. This should be mentioned in the section limitations. During the period 2005-2015 the treatment for coronary artery disease has changed a lot including the prescription pattern as well as the frequency and the intensity of statin treatment. There are numerous potential confounding factors.
Thank you for your comments, and we appreciate the reviewer’s comments. Indeed, one of the major limitations of this study is its retrospective nature and that patients were enrolled from a single institution. This has been added to the limitation section at in the Discussion on Lines 71-732 (marked red). We are also aware of the possible confounding effects of statin use on the association between CONUT scores and clinical outcomes of patients undergoing PCI. Thus, the effect of statin use has been adjusted using cox regression analysis in model 2 and model 3, which is shown in Table 3 (mentioned in the Discussion on Lines 76-79).
- As statin treatment improves CV outcome but also increases CONUT score, this interaction should be more explored by the authors. It is not clear to what extent the score is negatively influenced by prescribing statins.
Thank you for your comments and we appreciate the reviewer’s comments. We share the reviewers’ concern regarding interaction between statin use and CONUT scores during the CV events. We have added a new analysis in the subgroup analysis to investigate whether statin use affect the hazard ratio for future events in patients after PCI. We determined high CONUT scores remain independently associated with the risk of developing future CV events and there is no significant interaction between statin use and CONUT with regards to the risk of a future adverse event. We have revised our manuscript and revised the figures (Discussion Lines 79-81, supplementary figures 1-5) to include this analysis.
- The authors should also investigate whether there is a supplementary value of the CONUT score (which reflects not only nutrition but also inflammation and immune response) in comparison to a predominantly nutritional value such as albumin.
Thank you for your comment. Albumin levels are not solely related to nutritional status. It is also related to the acute phase reaction. Lower albumin levels may be a marker of persistent injury to arteries and progression of atherosclerosis and thrombosis. To address these possibilities, we have compared the predictive value for albumin and the CONUT score by C-index. This comparison revealed a trend toward higher C-index values in the CONUT score group, although we found no significant difference in the prediction of cardiac death (0.5913 vs 0.6520, p=0.9984), CHF (0.6246 vs 0.6498, p=0.9685), CV death (0.7032 vs 0.7238, p=0.818), MACE (0.6265 vs 0.6608, p=0.9913) and total CV events (0.5648 vs 0.5838, p=0.9848) respectively. Nevertheless, the CONUT score comprises of three compounds of which, as a whole, could better reflect the status of CAD patients. We have revised our manuscript with this in mind and thank you for your comments.
- Furthermore Figure 2 is impossible to read.
Thank you for your comment. We attempted to improve the figure resolution but did not succeed because merging 5 individual figures is too crowded. We’ve decided to separate Figure 2 into 5 individual supplementary figures to make these figures more readable.
- Finally, the conclusion is grammatically not correct.
Thank you for this reminder. The conclusion has been rephrased in the Discussion at Lines 66-68 (marked in red): This study shows that CONUT score is an independent predictor of long-term cardiovascular disease to identify outcomes in patients with coronary artery disease, including those presenting with acute coronary syndromes.
Reviewer 2 Report
This is a single centre retrospective study of nutrition status (measured by CONUT score) as a predictor of MACE/adverse cardiovascular events in patients with symptomatic CAD who underwent PCI from 2005-2015 with a mean follow up of 58.5 months. The major findings were that CONUT score was significantly associated with major adverse cardiac events and death, independent of other predictors and also added to a baseline risk predictor model. This is interesting and novel.
Major issues:
Study population is patients with symptomatic CAD who underwent PCI from 2005-2015. However, the authors then comment on definition of MI and ischaemia. Was the eligibility ALL consecutive CAD patients treated with PCI, or did patients have to have an MI or a positive stress test. This is confusing.
Methods state that “All study patients were followed up regularly after enrollment.” However, as a restrospective study – HOW were patients followed up: this is not clear. There will clearly be a bias or lack of follow up if this was done retrospectively. Instead the authors simply state the “protocol for CV event follow-up was similar as previously reported [14,15].”BUT these two papers are NOT similar to the current study – the first was the Biosignature study : a nationwide prospective cohort study to identify risk factors among CHD patients in stable condition at baselineacross 8 centres. The second study was Between March 1998 and June 2001, with 1000 consecutive patients with angina pectoris sent to a single catheterization room for coronary angiogram in a medical center evaluated. This aspect of the methods that is critical to evaluating the robustness of the findings is missing.
The medication use in Table 1: only 50% overall and 30% of those with severe CONUT scores are on statins. These patietns just received PCI for CAD – statin use is usually >90% in this cohort. When was this medication use taken from? Similarly, antiplatelet use is 75%, in patients receiving PCI – this has not specific aspirin versus DAPT. Why is this % so low?
Reducing LDL is seen in those with poorer nutrition scores on the CONUT score – how is this reconciled with reduction of LDL for secondary prevention through statins to reduce risk of future events? This is not discussed apart from this sentence: “Cholesterol homeostasis occurs as part of innate immune responses and may augment inflammatory responses, leading to atherosclerosis[22,23].” This should be expanded on.
Is the CONUT score gendered? As women would have lower LDL /TC / weight
Mean Stent number also higher in cohorts with poorer nutrition – hence likely reflective of more extensive CAD – is this factored in? Do we know the rate of single versus triple verssel disease, of complete versus incomplete revascularisatin. It is possible those who were severely malnourished had more extensive CAD / less complete revascularisation.
Minor suggestions:
Results – line 121 – put the centre in the methods, and remove from the results.
Figure 1: all the x-axis and the legend should state “Cumulative survival FREE of CV death/MI etc” In addition, the years on the y axis should be truncated – it is not appropriate to show the KM out to 12 years when even by 10 years there are 0 patients in the severe category and VERY small number in th other categories. Would be better to cull at 6 years or so.
Multiple Grammatical errors as outlined below:
ABSTRACT: conclusions:
“CONUT score improved the risk prediction of adverse event compared to traditionalrisk factors in CAD patients after PCI” should be ‘adverse events’
Introduction:
‘Therefore, we aim to evaluate the prognostic value of the CONUT score in patients with coronary artery diseases undergoing percutaneous coronary intervention (PCI).” Should be ‘coronary artery disease’
Methods line 76
“and has used for nutrition status evaluation” should be ‘and has BEEN used’
Line 82 should be “Primary endpoint wasmajor cardiovascular events”
Line 121/2 – “The baseline characteristics of participants according TO COUNT score categories ARE shown in Table 1.”
Line 124 “were older, HAD lower body weight”
Line 125 same again “HAD” not have
“Figure 1 shows the showed th” – should be ‘shows the…’
Discussion line 11 “Nutrition status playS an important role in maintaining vital organ function in THE human body[17,18].”
Line 14/14 “the evidence of malnutrition to cardiovascular disease mainly focused on CHF
15 population.” Should be “the correlation of malnutrition with cardiovascular disease has mainly focused on CHF”
Line 15/16 Our previous has demonstrated that low serum albumin concentration worsens
16 prognosis of patients with stable CAD[14], suggesting the important role of nutrition status in long17
term outcome in CAD patients
Should be ‘our previous STUDY’
Line 23/25 These elements contributes to pathophysiology of coronary artery disease[25]. This makes the CONUT score a reasonable tool to determine the nutritional status of CAD population.
Line 39 leading to the occurrence of adverse outcomeS
Conclusions line 62-64 This study CONUT score as an independent predictor of long-term cardiovascular to identify outcomes in patients with coronary artery disease, including those presenting with acute coronary
syndromes.
Author Response
Major issues:
- Study population is patients with symptomatic CAD who underwent PCI from 2005-2015. However, the authors then comment on definition of MI and ischaemia. Was the eligibility ALL consecutive CAD patients treated with PCI, or did patients have to have an MI or a positive stress test. This is confusing.
Thank you for pointing out the confusing statement regarding the study population. CAD was diagnosed by at least one of the following modalities: (1) a history of myocardial infarction as evidenced by ischemic change in 12-lead electrocardiography (ECG) and elevated cardiac enzymes; (2) a history of angina with ischemic ECG changes, positive response to stress test, or presence of significant stenotic lesion in coronary computed tomography angiography (CCTA). CAD patients who fulfilled the above criteria and received coronary intervention (percutaneous coronary intervention (PCI) with either coronary stenting or balloon angioplasty) were enrolled. We have revised our manuscript and thank you for your comments (Method section Lines 60-65).
- Methods state that “All study patients were followed up regularly after enrollment.” However, as a retrospective study – HOW were patients followed up: this is not clear. There will clearly be a bias or lack of follow up if this was done retrospectively. Instead the authors simply state the “protocol for CV event follow-up was similar as previously reported [14,15].”BUT these two papers are NOT similar to the current study – the first was the Biosignature study : a nationwide prospective cohort study to identify risk factors among CHD patients in stable condition at baseline across 8 centres. The second study was Between March 1998 and June 2001, with 1000 consecutive patients with angina pectoris sent to a single catheterization room for coronary angiogram in a medical center evaluated. This aspect of the methods that is critical to evaluating the robustness of the findings is missing.
Thank you for your comment, and we appreciate the reviewer’s comments. We have added the method describing how our patients were followed-up: The study patients resumed to outpatient clinic visits within two to four weeks after discharge. After their first return visit, they were then regularly followed up at one- to three- month intervals. Data for follow-up were retrospectively obtained from hospital records and chart reviews (Lines 82-84, marked in red). To avoid confusion, we changed two references cited and we appreciate this comment.
- Yang, Y.L.; Wu, C.H.; Hsu, P.F.; Chen, S.C.; Huang, S.S.; Chan, W.L.; Lin, S.J.; Chou, C.Y.; Chen, J.W.; Ju-Pin, P., et al. Systemic immune-inflammation index (SII) Predicted Clinical Outcome in Patients With Coronary Artery Disease. European journal of clinical investigation 2020, 10.1111/eci.13230, e13230, doi:10.1111/eci.13230.
- Lim, S.S.; Yang, Y.L.; Chen, S.C.; Wu, C.H.; Huang, S.S.; Chan, W.L.; Lin, S.J.; Chen, J.W.; Chou, C.Y.; Pan, J.P., et al. Association of variability in uric acid and future clinical outcomes of patient with coronary artery disease undergoing percutaneous coronary intervention. Atherosclerosis 2020, 297, 40-46, doi:10.1016/j.atherosclerosis.2020.01.025.
- The medication use in Table 1: only 50% overall and 30% of those with severe CONUT scores are on statins. These patients just received PCI for CAD – statin use is usually >90% in this cohort. When was this medication use taken from? Similarly, antiplatelet use is 75%, in patients receiving PCI – this has not specific aspirin versus DAPT. Why is this % so low?
Thank you for your comments, and we apologize for the typo in writing. The “anti-platelet” should be “aspirin” and we have added the percentage of “P2Y12 inhibitor” information. As to why statin prescription rate is not more than 90%, there are several possible reasons. First, this is a retrospectively database collection of data from 2005 to 2015. During the period between 2005-2015 the treatment for coronary artery disease has changed significantly and statin prescription rate at that time was not so high as our current guideline. Second, patients in the highest CONUT score category have a lower baseline LDL than other categories identified in our current study and that is why there are relatively low statin prescription rates in the severe CONUT group. A similar observation of lipid paradox has been mentioned previously and malnutrition with increased inflammation status may be responsible for this observation. Therefore, we adjusted statin use as well as LDL and HDL values in the Cox model and CONUT was still independently associated with the risk of future adverse event in patients after PCI. We have revised our manuscript and added this description in our study limitations section.
- Reducing LDL is seen in those with poorer nutrition scores on the CONUT score – how is this reconciled with reduction of LDL for secondary prevention through statins to reduce risk of future events? This is not discussed apart from this sentence: “Cholesterol homeostasis occurs as part of innate immune responses and may augment inflammatory responses, leading to atherosclerosis[22,23].” This should be expanded on.
Thank you for this comment. The link between elevated serum cholesterol levels and cardiovascular disease was confirmed with the discoveries of the low-density lipoprotein (LDL) receptor and the statins treatment. However, it is interesting to note a reverse association between LDL values and the CONUT scores were found in this study. The lipid paradox has been reported in several clinical studies including myocardial infarction and heart failure patients, though the cause of lipid paradox remains unknown. Possible explanations are that these patients have high vascular inflammation status even though a LDL value that is not very high. Low lipid levels may be markers for advanced disease and systemic inflammatory activation, and could be reflect malnutrition and cachexia, which are known to be associated with increased mortality in various chronic diseases. Lower LDL levels may be associated with poor nutritional status, making patients more vulnerable to future events. Second, these patients may not receive aggressive statin therapy because of their lower baseline LDL levels, and relatively low statin use may be responsible for the higher risk for being in the high CONUT score group. To explore this issue, our current study adjusted confounding factors, which include statin use, LDL and HDL values and higher CONUT scores were still found to independently associate with the risk for a future adverse event in CAD patients after PCI. We have added this description in our study limitation in the section of Discussion Lines 24-38.
- Is the CONUT score gendered? As women would have lower LDL /TC / weight
Thank you for your comment. Indeed, women tend to have lower LDL/TC/weight. Previous studies have not implemented different categorization of the CONUT score based on gender. To address this question, we have calculated the mean CONUT score in males (n=2542) and females (n=576), and the scores were 2.59 ± 2.21 and 2.66 ± 2.40 respectively, with a p-value equals to 0.515.
- Mean Stent number also higher in cohorts with poorer nutrition – hence likely reflective of more extensive CAD – is this factored in? Do we know the rate of single versus triple vessel disease, of complete versus incomplete revascularization. It is possible those who were severely malnourished had more extensive CAD / less complete revascularization.
Thank you for your comment. The rate of different CAD severity has been added to Table 1 in terms of number of culprit vessel. In the severely malnourished group, 57.6% of patients had triple vessel disease. Those with poorer nutrition had more extensive CAD (Results section LLines 129-131, marked in red), therefore we expect poorer outcomes in those patients.
Minor suggestions:
Results – line 121 – put the centre in the methods, and remove from the results.
Thank you for your suggestion. The centre has been moved to line 59 in the method section, marked in red.
Figure 1: all the x-axis and the legend should state “Cumulative survival FREE of CV death/MI etc” In addition, the years on the y axis should be truncated – it is not appropriate to show the KM out to 12 years when even by 10 years there are 0 patients in the severe category and VERY small number in th other categories. Would be better to cull at 6 years or so.
Thank you for your comment. We have updated the figure by correcting the figure legend, and cut the KM to 10 years.
Multiple Grammatical errors as outlined below:
ABSTRACT: conclusions:
“CONUT score improved the risk prediction of adverse event compared to traditional risk factors in CAD patients after PCI” should be ‘adverse events’
Thank you for your reminder. The grammatical mistake at line 33 has been corrected.
Introduction:
‘Therefore, we aim to evaluate the prognostic value of the CONUT score in patients with coronary artery diseases undergoing percutaneous coronary intervention (PCI).” Should be ‘coronary artery disease’
Thank you for your reminder. The grammatical mistake at line 55 has been corrected.
Methods line 76
“and has used for nutrition status evaluation” should be ‘and has BEEN used’
Thank you for your reminder. The mistake at line 77 has been corrected.
Line 82 should be “Primary endpoint was major cardiovascular events”
Thank you for your reminder. The grammatical mistake at line 85 has been corrected.
Line 121/2 – “The baseline characteristics of participants according TO COUNT score categories ARE shown in Table 1.”
Thank you for your reminder. The grammatical mistake at line 126-127 has been corrected.
Line 124 “were older, HAD lower body weight”
Thank you for your reminder. The grammatical mistake at line 128 has been corrected.
Line 125 same again “HAD” not have
Thank you for your reminder. The grammatical mistake at line 130 has been corrected.
“Figure 1 shows the showed th” – should be ‘shows the…’
Thank you for your reminder. The grammatical mistake at line 140 has been corrected.
Discussion line 11 “Nutrition status playS an important role in maintaining vital organ function in THE human body[17,18].”
Thank you for your reminder. The grammatical mistakes at Discussion line 11 have been corrected.
Line 14/14 “the evidence of malnutrition to cardiovascular disease mainly focused on CHF population.” Should be “the correlation of malnutrition with cardiovascular disease has mainly focused on CHF”
Thank you for your reminder. The sentence of Discussion line 14 has been rephrased as suggested.
Line 15/16 Our previous has demonstrated that low serum albumin concentration worsens prognosis of patients with stable CAD[14], suggesting the important role of nutrition status in long17 term outcome in CAD patients
Should be ‘our previous STUDY’
Thank you for your reminder. The sentence of Discussion line 15/16 has been corrected.
Line 23/25 These elements contributes to pathophysiology of coronary artery disease[25]. This makes the CONUT score a reasonable tool to determine the nutritional status of CAD population.
Thank you for your reminder. The grammatical mistake has been corrected.
Line 39 leading to the occurrence of adverse outcomeS
Thank you for your reminder. The grammatical mistake at Discussion lines 42-43 has been corrected.
Conclusions line 62-64 This study CONUT score as an independent predictor of long-term cardiovascular disease to identify outcomes in patients with coronary artery disease, including those presenting with acute coronary syndromes.
Thank you for your reminder. The conclusion has been rephrased at Discussion Lines 87-89 (marked in red): This study shows that CONUT score is an independent predictor of long-term cardiovascular to identify outcomes in patients with coronary artery disease, including those presenting with acute coronary syndromes.
Round 2
Reviewer 2 Report
Thank you for the responses to the suggestions.
One more correction:
page 2, line 62
the presence of a significant stenotic lesion ON coronary computed tomography angiography
Author Response
page 2, line 62
the presence of a significant stenotic lesion ON coronary computed tomography angiography
Response: Thank you for your reminder. The mistake has been corrected on page 2, line 62: ...the presence of a significant stenotic lesion on coronary computed tomography angiography (CCTA).